# MolNetEnhancer: Enhanced Molecular Networks by Integrating Metabolome Mining and Annotation Tools

**DOI:** 10.3390/metabo9070144

**Published:** 2019-07-16

**Authors:** Madeleine Ernst, Kyo Bin Kang, Andrés Mauricio Caraballo-Rodríguez, Louis-Felix Nothias, Joe Wandy, Christopher Chen, Mingxun Wang, Simon Rogers, Marnix H. Medema, Pieter C. Dorrestein, Justin J.J. van der Hooft

**Affiliations:** 1Collaborative Mass Spectrometry Innovation Center, Skaggs School of Pharmacy and Pharmaceutical Sciences, University of California San Diego, La Jolla, CA 92093, USA; 2Department of Congenital Disorders, Center for Newborn Screening, Statens Serum Institut, 2300 Copenhagen, Denmark; 3Research Institute of Pharmaceutical Sciences, College of Pharmacy, Sookmyung Women’s University, Seoul 04310, Korea; 4Glasgow Polyomics, University of Glasgow, Glasgow G12 8QQ, UK; 5School of Computing Science, University of Glasgow, Glasgow G12 8QQ, UK; 6Bioinformatics Group, Department of Plant Sciences, Wageningen University, 6708 PB Wageningen, The Netherlands; 7Department of Pediatrics, University of California San Diego, La Jolla, CA 92093, USA; 8Center for Microbiome Innovation, University of California San Diego, La Jolla, CA 92093, USA

**Keywords:** chemical classification, in silico workflows, metabolite annotation, metabolite identification, metabolome mining, molecular families, networking, substructures

## Abstract

Metabolomics has started to embrace computational approaches for chemical interpretation of large data sets. Yet, metabolite annotation remains a key challenge. Recently, molecular networking and MS2LDA emerged as molecular mining tools that find molecular families and substructures in mass spectrometry fragmentation data. Moreover, in silico annotation tools obtain and rank candidate molecules for fragmentation spectra. Ideally, all structural information obtained and inferred from these computational tools could be combined to increase the resulting chemical insight one can obtain from a data set. However, integration is currently hampered as each tool has its own output format and efficient matching of data across these tools is lacking. Here, we introduce MolNetEnhancer, a workflow that combines the outputs from molecular networking, MS2LDA, in silico annotation tools (such as Network Annotation Propagation or DEREPLICATOR), and the automated chemical classification through ClassyFire to provide a more comprehensive chemical overview of metabolomics data whilst at the same time illuminating structural details for each fragmentation spectrum. We present examples from four plant and bacterial case studies and show how MolNetEnhancer enables the chemical annotation, visualization, and discovery of the subtle substructural diversity within molecular families. We conclude that MolNetEnhancer is a useful tool that greatly assists the metabolomics researcher in deciphering the metabolome through combination of multiple independent in silico pipelines.

## 1. Introduction

Metabolomics has matured into a research field generating increasing amounts of metabolome profiles of complex metabolite mixtures aiming to provide biochemical insights. Mass spectrometry has become the workhorse of metabolomics and typical untargeted experiments currently result in qualitative and semiquantitative information on several thousands of molecular ions across tens to hundreds of samples. Technical advances in the last decade have allowed researchers to fragment increasing amounts of mass peaks that result in mass fragmentation spectra (MS/MS or MS2). Metabolite annotation and identification tools have benefited from these advances as now more MS2 spectra per sample can be queried in reference libraries in order to find candidate structures or submitted to in silico tools that propose a putative structure [1,2,3,4,5,6,7,8,9].

Despite these tremendous advances, a key challenge remaining for metabolomics researchers is to biochemically interpret large-scale untargeted metabolomics studies due to the complexity of the metabolomes represented by mass fragmentation spectra to which actual chemical structures need to be assigned, and for which reference spectra are not available. In biological samples, many metabolites share molecular substructures and form structurally related molecular families (MFs) of various chemical classes, which has inspired metabolome mining tools exploiting these biochemical relationships. Based on the assumption that structurally similar molecules (analogs) generate similar mass spectrometry fragmentation spectra, one can group analogs by comparing their fragmentation spectra resulting in the construction of molecular families. To do this on a larger scale, computational tools have been developed such as molecular networking (MN) [7]. However, to actually annotate structural information additional sources are usually needed such as library matches, candidate structures from libraries or chemical class annotations.

Indeed, since the molecular networking approach was proposed in 2012 [10], numerous complementary metabolome mining workflows as well as annotation and classification tools have been introduced including SIRIUS [3], CSI:FingerID [4], MetFusion [11], MetFamily [12], and many others of which some also use molecular networks as basis [1,2,7,8,13,14,15,16,17,18,19,20,21,22,23,24] and their combined use for natural product discovery was very recently reviewed [25]. Where tandem mass spectral molecular networking efficiently can group molecular features in molecular families [10], MS2LDA can discover substructures, not only based on common fragment peaks but also common neutral losses, which can aid in further annotation of subfamilies and shared modifications [14]. These metabolome mining tools typically take MS/MS spectra as input, such as the open formats Mascott Generic Format (MGF), the mzML, or mzXML format, and generate tables where a fragmented mass feature is linked to other fragmented mass features or substructure patterns. Reference fragmentation spectra in public repositories are still very few. Thus, on average only 2–5% percent of MS2 spectra acquired in a typical LC–MS/MS experiment can be matched to known molecules [26]. Complementary to library matching, in silico tools such as Network Annotation Propagation (NAP) [8], DEREPLICATOR [1], VarQuest [2], or SIRIUS+CSI:FingerID [4] predict fragmentation spectra in silico from known structures and allow for effective searching in chemical databases for candidate structures. These metabolome annotation tools also take MS/MS spectra as input and typically use precursor masses to find candidate structures in compound databases followed by a ranking of those structures based on the similarity of the predicted and experimental MS/MS data. The output is typically a table with candidate structures found for each mass feature and associated score. These tools typically differ in the compound databases they use to query for candidate structures, or the processing of mass spectrometry data. For example, SIRIUS+CSI:FingerID first builds annotated fragmentation trees before searching molecular structures in large compound databases. DEREPLICATOR and VarQuest are annotation tools that match structures from a large database of Peptidic Natural Products to MS/MS spectra, whereby DEREPLICATOR looks for exact matches and VarQuest also allows for one modified amino acid. It is important to realize that each tool has its own set of parameters that will affect the number of annotated features.

The outputted structural information for each mass feature can be mapped on a molecular network, for example, to show for which mass features library matches or in silico predicted structural matches are available. The recently introduced Network Annotation Propagation (NAP) also exploits the network topology to rerank candidate structure lists based on neighboring matches within molecular families [8]. Furthermore, when using multiple annotation tools, the structural information they provide may support each other increasing confidence in the annotation.

To assess whether molecular families are of particular interest for your research question, knowing their chemical class may provide sufficient information. The recently proposed ClassyFire tool [16] takes molecular descriptors as SMILES or InchiKeys as input and outputs hierarchical chemical ontology terms. Thus, the candidate structures outputted for each mass feature by the metabolome annotation tools mentioned above can now be automatically chemically classified. When that is done at larger scale for an entire molecular family, one can combine those chemical class terms and assess whether particular terms are enriched.

Taken together, all these recent developments enable the discovery of relations between millions of spectra and the listing of candidate structures from various spectral libraries or alternatively from compound libraries using in silico approaches.

Whilst each of those tools produce useful structural information, their combined application has been hampered by the use of different file formats, platforms, and the challenge to match molecular features across the outputs of these tools. We postulate that whilst each tool provides complementary insights, their combined use allows an increased level of biochemical interpretation, i.e., the sum becomes greater than the individual parts. Furthermore, it would be practically advantageous to combine all these results in one place. We have previously described the integration of Mass2Motifs and chemical classifications with molecular networks to assess the chemical diversity within a subset of species of the plant genus *Euphorbia* [27] and the plant family Rhamnaceae [28]. However, in those studies, integration was achieved using custom in-house scripts in R, hampering adoption by the community. Moreover, the results of the peptide annotation tools DEREPLICATOR and VarQuest were not included in those custom scripts.

Here, we introduce MolNetEnhancer a software package available in Python and R that unites the output of many of the above-mentioned metabolome mining and annotation tools (GNPS molecular networking, MS2LDA substructure discovery, and in silico annotation tools) independent of what dataset it processes, thus making the algorithm accessible in an easy-to-use format to the community (Figure 1). MolNetEnhancer discovers molecular families (MFs), subfamilies, and subtle structural differences between family members. The workflow enhances the currently available molecular networking methods based on either MS-Cluster [29] (classical) or MZmine2 [30] (also called “feature-based molecular networking”) and results in annotated molecular networks that can be explored in Cytoscape [31]. We applied MolNetEnhancer to publicly available mass spectrometry fragmentation data ranging from marine-sediment and nematode-related bacteria, to *Euphorbia* and Rhamnaceae plants. Illustrated by four case studies, we demonstrate how our integrative workflow discovers dozens of MFs in large-scale metabolomics studies of these plant and bacterial extracts. Moreover, discovered MFs can be divided into subfamilies using the mapped MS2LDA results. Structural annotation of Mass2Motifs is facilitated by having chemical and structural annotations at hand, for example by recognizing substructures in peptidic molecules. We conclude that our workflow provides chemical refinement of metabolomics results beyond spectral matches through large-scale MF and substructure discovery and annotation by integrating outputs of various tools in one place allowing for enhanced visualization. This also guides the metabolomics researcher in prioritizing MFs to explore and in structurally annotating molecules.

## 2. Materials and Methods

MolNetEnhancer is a software package available in Python and R that unites the output of several metabolome mining and annotation tools, including mass spectral molecular networking through GNPS, unsupervised substructure discovery through MS2LDA and in silico structure annotation, for example through NAP, DEREPLICATOR, or SIRIUS+CSI:FingerID (Figure 1). Before using the MolNetEnhancer workflow, the user will run each metabolome mining tool separately:Perform mass spectral molecular networking analysis through the Global Natural Products Social Molecular Networking platform (https://gnps.ucsd.edu).Perform in silico chemical structural annotation using for example Network Annotation Propagation (NAP) and DEREPLICATOR through the GNPS platform. Alternatively, other in silico tools for putative chemical structural annotation (e.g., SIRIUS+CSI:FingerID) [3,4] can also be used.Perform unsupervised substructure discovery using MS2LDA (http://ms2lda.org).

For documentation of steps 1–3 the user is referred to the original publications and guidelines for each tool [1,2,7,8,14]. Section 8 contains links to tutorials of the analysis tools used in this study. Functions implemented in the MolNetEnhancer workflow can then be used to combine the outputs created in step 1–3 such that

aSubstructure information retrieved through MS2LDA is integrated with mass spectral molecular networks.bMost abundant chemical classes per molecular family are retrieved based on GNPS structural library hits and in silico chemical structural annotation and integrated within the mass spectral molecular networks.

MolNetEnhancer is freely available on GitHub at https://github.com/madeleineernst/pyMolNetEnhancer and https://github.com/madeleineernst/RMolNetEnhancer. Interactive Jupyter example notebooks and a step by step tutorial guide the user to build enhanced mass spectral molecular networks, which are outputted in the graphml format for visualization in Cytoscape.

Currently, two distinct methods from raw data to MNs exist. One method takes all MS2 spectra found in the input files and uses MS-Cluster to prepare a set of representative “consensus” MS2 spectra for molecular networking, and the other method uses MZmine2 for data preprocessing, which performs molecular feature detection at the MS1 level and associates each MS1 feature with its respective MS2 spectra to send off to GNPS Molecular Networking. The here proposed MolNetEnhancer workflow can enrich both these molecular networking methods with Mass2Motif presence and chemical class annotations.

Substructural information retrieved through MS2LDA is integrated in two ways within the mass spectral molecular networks. Shared substructures or motifs between two molecular features are visualized as multiple edges connecting the nodes. Furthermore, motifs found within a molecular feature can be visualized as pie charts, where the relative abundance of each motif represents the overlap score, a score measuring how much of the motif is present in the spectrum [32]. Furthermore, for each molecular family, the x most shared motifs are shown, where x is defined by the user. An example of such a molecular family with motifs mapped is shown in Figure 6 in the results section.

To retrieve the most abundant chemical classes per molecular family, all chemical structures obtained through GNPS library matching, and in silico chemical structural annotation are submitted to automated chemical classification and taxonomy structure using ClassyFire [16]. This retrieves chemical classes for each of the putative structures submitted organized in five hierarchical levels of a chemical taxonomy (kingdom, superclass, class, subclass, and direct parent). For each level of the chemical ontology, a score is calculated, which represents the most abundant chemical class found for the structural matches within the molecular family. It is important to note that a high score does not represent a higher confidence in the true identity of the chemical structures found within the molecular family, but indicates more consistency as more structural matches obtained for this molecular family fall within the same chemical class. Figure 2 exemplifies how the score is calculated. Given a molecular family consisting of six molecular features (nodes), the percentage of nodes classified as cinnamaldehydes, coumarins and derivatives, flavonoids and macrolactames at the chemical class level respectively is calculated. Each molecular feature can have multiple structural matches with multiple (e.g., node 2) or identical (e.g., node 3) chemical classes. A majority of the structural matches obtained in the network shown in Figure 2 were classified as flavonoids (2.25 out of six nodes), thus the molecular family is classified as flavonoids with a chemical classification score at the class level of 0.375 (2.25/6). For single nodes (molecular features which show no spectral similarity with any other molecular features found in the dataset) the chemical classes are retrieved analogously, however, it should be noted that single nodes often result in a very high score, as only one structural match is retrieved, corresponding to a score of 1 (1 node out of 1).

## 3. Results

### 3.1. MolNetEnhancer Workflow

MolNetEnhancer requires inputs from independent metabolome mining tools including mass spectral molecular networking through GNPS, unsupervised substructure discovery through MS2LDA and in silico structure annotation, for example through NAP, DEREPLICATOR or SIRIUS+CSI:FingerID (Figure 1). Provided with these inputs, MolNetEnhancer consists of two independent steps. During the first step, molecular substructures detectable by co-occurring fragment ions or neutral losses, so called Mass2Motifs, are mapped onto a Molecular Network. Each node in the network represents a molecular feature, whereas Mass2Motifs represent substructural features. Most fragmented mass peaks (precursor ions) represent molecular ions, although fragmented mass peaks may also represent adducts of one and the same molecule, in source fragments or doubly-charged peaks [33]. For simplicity, we will refer to any fragmented mass peak as molecular feature throughout the manuscript. Mass2Motifs contained within each molecular feature can be visualized as pie charts on the nodes. Alternatively, Mass2Motifs shared across multiple molecular features can be visualized as multiple lines (edges) connecting the nodes. In a second step, most abundant chemical classes per molecular family based on candidate structures from in silico annotation tools as well as GNPS library matches can be mapped through chemical classification using ClassyFire [16]. A chemical classification score is calculated representing what percentage of nodes within a molecular family are attributed to a given chemical class (see Section 2 and Figure 2 therein). In Section 3.2, Section 3.3, Section 3.4 and Section 3.5 we show how MolNetEnhancer can accelerate and enrich chemical information retrieval in 4 case studies, comprising two plant and two bacterial publicly accessible datasets. The MolNetEnhancer workflow results in one graphml network file that contains all the structural information obtained from the individual tools. Such a file can be easily imported into network visualization tools such as Cytoscape [31], an environment where additional metadata on the molecular features can be added. In addition, all structural information is also available as tab delimited text files.

### 3.2. Case Study 1: Annotation of Euphorbia Specialized Metabolites Using MolNetEnhancer

With more than 2000 species worldwide, the plant genus *Euphorbia* is among the most species-rich and diverse flowering plants on earth [34,35]. Besides exhibiting an extreme diversity in its growth forms and habitat types, the genus has also attracted interest within natural products drug discovery [36,37]. *Euphorbia* species are chemically highly diverse, particularly within macro- and polycyclic diterpenoids, biosynthetically derived from a head-to-tail cyclization of the tetraprenyl pyrophosphate precursor, which have been found to exhibit a range of biological activities with pharmaceutical interest, such as antitumor, antimicrobial or immunomodulatory activity [36]. Ingenol mebutate for example, a diterpenoid originally isolated from *Euphorbia peplus* L. is marketed for the topical treatment of actinic keratosis, a precancerous skin condition [38], however production through plant extraction or chemical synthesis is inefficient and expensive [39,40].

A key interest is therefore to find species within the genus producing higher quantities of ingenol mebutate or other close diterpenoid analogs exhibiting biological activities with pharmaceutical interest. We have previously assessed chemical diversity within a representative subset of species of the plant genus *Euphorbia* [27]. A major challenge is the rapid identification of known and unknown *Euphorbia* diterpenoid structures. Using MolNetEnhancer, we were able to significantly accelerate manual annotation of diterpenoids and retrieve chemical structural information, even for molecular families with no structural matches in the GNPS spectral libraries.

An example of how MolNetEnhancer increases chemical structural information throughout two molecular families is highlighted in Figure 3. Using GNPS spectral library matching, chemical structural information for only one molecular feature was obtained, and manual propagation of the annotation throughout molecular family (i) was limited given that the annotated ion exhibited one neighbor only. No structural information could be retrieved for family (ii), where no chemical structural information was retrieved through GNPS library matching (Figure 3a).

Using MolNetEnhancer however, we were able to highlight substructural Mass2Motifs within both molecular families (Figure 3b). Substructural Mass2Motifs, putatively annotated as a *Euphorbia* diterpenoid backbone skeleton with mass peaks at *m*/*z* 313, 295, and 285 were found both in molecular families (i) and (ii) (Figure 3b). Manual annotation of these Mass2Motifs was possible by comparing mass fragments of the library spectrum to mass fragments contained in the Mass2Motifs. A mirror plot comparing the GNPS reference spectrum to the unknown spectrum found in our samples is shown in Appendix A. The exact *Euphorbia* backbone skeleton type could not be identified unambiguously, as many *Euphorbia* diterpenoid skeletons are isomeric and their respective MS2 spectra are identical or very similar. A *Euphorbia* backbone skeleton with masses at *m*/*z* 313, 295, 285 can either result from a jatrophane, deoxy tigliane, or ingenane ester like skeleton [41,42]. Furthermore, we were able to see that molecular family (ii) contains substructural Mass2Motifs related to a nicotinoyl side chain. Manual annotation of these Mass2Motifs was possible by comparing chemical structures retrieved through NAP in silico structure annotation with mass fragments found in the Mass2Motifs. Motifs 432 and 180 were both found to contain mass peaks at *m*/*z* 106 and 124, possibly resulting from a nicotinoyl side chain and a hydroxylation (Figure 3b). Chemical structures retrieved through in silico annotation or library matching can aid the manual annotation of Mass2Motifs and vice versa annotated Mass2Motifs can aid the propagation of chemical structural information throughout the network. Additionally, chemical structural hypotheses can be reinforced by taking into consideration both substructural information as well as chemical class information obtained through in silico annotation and library matching. Most chemical structures retrieved for molecular family (i) and (ii) were diterpenoids of the jatrophane, tigliane or ingenane type and substructures related to these *Euphorbia* diterpenoid backbone skeletons were also found within the Mass2Motifs (Figure 3c).

In conclusion, using MolNetEnhancer we were able to significantly increase chemical structural annotations obtained from retrieving chemical structural information of one molecular feature through GNPS library matching (Figure 3a), to retrieving chemical structural information at an annotation level 3 (putatively characterized compound classes) according to the Metabolomics Standard Initiative’s reporting standards [43] of two molecular families comprising 73 molecular features (Figure 3b–d). Finally, this information allowed us to conclude that within the investigated subset of molecular families *Euphorbia* diterpenoid skeletons of the jatrophane, deoxy tigliane, or ingenane ester type are found within all *Euphorbia* subgeneric clades, whereas nicotinoyl sidechain modifications are unique to subgenus *Esula* (Figure 3d).

### 3.3. Case Study 2: Annotation of Rhamnaceae Specialized Metabolites

Another case where we demonstrate the efficiency of MolNetEnhancer for enhancing the chemical annotation of metabolomics data is our previous study on the plant family Rhamnaceae [28]. Rhamnaceae is a cosmopolitan family including about 900 species, and Rhamnaceae species are known for their exceptional morphological and genetic diversity, which are thought to be caused by the wide geographic distribution and different habitats [44]. We applied an MS2-based untargeted metabolomics approach to get insights on the metabolomic diversity of this highly-diversified family, and MolNetEnhancer was used as a key to provide fundamental annotations for MS2 spectra.

As shown in Figure 4a, MolNetEnhancer provided the putative chemical classification of molecular families within the Rhamnaceae molecular network. After combining this chemical class annotations with taxonomic information of each molecular feature, the normalized distribution pattern of different classes of metabolites were analyzed. This revealed that the taxonomic clade Rhamnoid exhibits more diversified flavonoids, carbohydrates, and anthraquinones, while the Ziziphoid clade produces various triterpenoids and triterpenoid glycosides [28].

MolNetEnhancer allowed us to visualize and discover the subtle substructural diversity within the molecular families. In the molecular family of triterpenoid esters, for example, substructural differences of phenolic moieties such as protocatchuate, vanillate, and coumarate were easily recognized by analyzing the distribution of Mass2Motifs 28, 117, 120, and 191 (Figure 4b). Two flavonoid aglycone substructures, kaempferol and quercetin, were also distinguished by analyzing the distribution of Mass2Motifs 86, 130, and 149 in the molecular family of flavone 3-*O*-glycosides (Figure 4c). Mass2Motif 130 contained mass peaks at *m*/*z* 284, 255, and 227, while Mass2Motifs 86 and 149 covered mass peaks at *m/z* 300, 271, and 255. These fragment ions are well-known as characteristic fragments of kaempferol 3-*O*-glycosides and quercetin 3-*O*-glycosides [45,46,47], so these Mass2Motifs could be easily annotated. This case study shows how MolNetEnhancer facilitates the interpretation process and our knowledge on MS2 fragmentation, previously mainly applied manually by experts.

### 3.4. Case Study 3: Large Chemical Diversity Uncovered by Annotating Specialized Metabolites in Marine Sediment Streptomyces and Salinispora Bacterial Extracts

The MolNetEnhancer workflow was also applied to bacterial data sets to gain more detailed insights into their chemical richness. Crüsemann and coworkers created a molecular network of extracts of the marine sediment bacteria *Salinispora* and *Streptomyces* that formed the basis for this case study [48]. Figure 5 displays the molecular network colored by the most prevalent chemical class annotations. Whilst we can observe that the bacteria also produce a structurally diverse arsenal of molecules, its composition is clearly different from that of the Rhamnaceae plants in Figure 4a. The most prevalent chemical class annotations are “Carboxylic acid and derivatives” and “Prenol lipids” with the first containing peptide-related molecules and the latter containing terpenoid molecules. Both these classes of molecules are known to be produced by *Salinispora* and *Streptomyces* bacteria. The chemical classification scores (see Section 2) for the ClassyFire class and kingdom terms are presented in Appendix A. These scores aid in assessing chemical novelty and also provide information on the consistency of the chemical class annotations of the structural candidates.

From the 5930 network nodes, we discovered 300 Mass2Motifs using MS2LDA. From those, we could annotate 40 with structural information at various levels of structural details gained from spectral matching with the GNPS libraries or from the in silico annotation tools NAP, DEREPLICATOR, and VarQuest. For example, we could annotate an amino sugar-related Mass2Motif with fragment ions related to two known N,N-dimethyl amino sugars present in known specialized molecules from the bacteria studied [48]: dimethylamino-β-d-xylo-hexopyranoside (rosamicin) and N,N-dimethyl-pyrrolosamine (lomaiviticin) which have overlapping fragment ions and are therefore characterized by the same Mass2Motif. With a frequency of more than 70 throughout the entire molecular network (using probability and overlap score thresholds of 0.1 and 0.3, respectively, for the molecular feature—Mass2Motif connections), the amino sugar Mass2Motif can be used as a handle to identify known and potential novel natural products throughout network. Indeed, the Mass2Motif was found in all members of the Rosamicin MF (Figure 6a) and the Lomaiviticin MF (Appendix A). Moreover, the same amino sugar-related Mass2Motif was also found in all members of two yet unknown MFs (Figure 6b, Appendix A). In addition, the Mass2Motif was also found in a number of singletons not connected to any MF, often in combination with Mass2Motif 66 as well like we see for the rosamicin-related MF. Mass2Motif 66 represents the presence of an *m*/*z* 116 fragment which is likely also generated by the dimethylated amino sugar; in fact it may point to the dimethylamino-β-D-xylo-hexopyranoside moiety or something very similar as this fragment is absent in spectra from the lomaiviticin MF which contains the different dimethylated amino sugar N,N-dimethyl-pyrrolosamine. In most singletons, no other Mass2Motifs were discovered that could provide clues on the complete structures of these molecules; however, given the presence of the amino sugar moiety they are likely natural products and not core metabolites or contaminants—something that we could not confidently state without using the MolNetEnhancer workflow.

Another MF displayed in Figure 6c did not return any GNPS library hits; however, all its members shared Mass2Motif 154. Due to its indicative fragment ions, we could annotate this Mass2Motif as tryptophan-related, indicating that all these molecules contain a tryptophan core structure. Based on their shared Mass2Motif, the masses of the molecular features, and their fragmentation patterns, with the help of MolNetEnhancer we could now tentatively annotate this MF as tryptophan-related containing molecules such as small peptides or N-acyltryptophans. Figure 6d shows the peptidic MF of actinomycin-related molecules. The annotation of this MF was guided by DEREPLICATOR and VarQuest annotations as well as the Mass2Motif that 10 of its members shared. We could annotate this Mass2Motif as the peptide lactone ring (depsipeptide moiety) present twice in actinomycins using reference data from literature [49]. The unique combination of four actinomycin-related mass fragments was only present in the 10 MF members, thereby reinforcing the DEREPLICATOR and VarQuest annotations.

Furthermore, mapping the Mass2Motifs on the molecular network means that we can more easily track neutral loss-based motifs such as the loss of an acetyloxy group that was only found in *Streptomyces* MFs. Moreover, inspection of the MFs without annotated chemical classes revealed that they contained some Mass2Motifs with relatively low frequency throughout the data set—something that could point to a unique substructure or scaffold possibly from a unique biosynthesis enzymatic function. For example, Mass2Motif 35 has a frequency of 43 and was present in all four members of the MF in Appendix A. It is a mass-fragment-based Mass2Motif and with masses of 142, 100, and 58 Da it could be related to a polyamine-like structural feature. Finally, the MF in Appendix A shares the two still unknown loss-based Mass2Motifs 250 and 261 that have frequencies of 26 and 50, respectively. These are examples of Mass2Motifs representing potential novel chemistry that can now be easily tracked in the molecular network.

### 3.5. Case Study 4: Annotating Peptidic Motifs in Peptide-Rich Xenorhabdus/Photorhabdus Extracts

*Xenorhabdus* and *Photorhabdus* are Gammaproteobacteria that live in symbiotic association with soil-dwelling nematodes of the genus *Steinernema* [50,51]. Eventually as a consequence thereof, they spend a large amount of their resources to the production of specialized metabolites, in particular nonribosomal peptides and polyketides. Tobias and coworkers recently published metabolomics data of 25 *Xenorhabdus* and five *Photorhabdus* strains to explore metabolic diversity amongst these strains [50]. Here, we applied MolNetEnhancer on this publicly available molecular networking data to further probe the chemical diversity previously found. The 6228 network nodes were analyzed with MS2LDA to discover 300 Mass2Motifs. Furthermore, we also submitted the *Xenorhabdus*/*Photorhabdus* molecular networking data to NAP, DEREPLICATOR, and VarQuest to run the MF chemical class annotation pipeline. By far the majority of the 46 annotated motifs were peptide, amino acid, or likely to be peptidic-related which fits with the ClassyFire predicted peptide-related MFs present in the *Xenorhabdus*/*Photorhabdus* extracts with “Carboxylic acids and derivatives” and “Peptidomimetics” as most frequently occurring annotations (see Figure 7, with corresponding chemical classification scores in Appendix A). We could also annotate an indole-related Mass2Motif which can be part of peptides/amino acids. An exception is the ethylphenyl-related Mass2Motif that was found in 478 molecules (out of 6228 nodes, corresponding to 7.7%) of the *Xenorhabdus*/*Photorhabdus* extracts. This can be explained by the reported production of phenylethylamides, dialkylresorcinoles, and cyclohexadions derivatives by the studied strains [52].

Annotations included Mass2Motifs that form peptidic substructures related to well-known *Xenorhabdus* peptidic families such as the commonly found bioactive rhabdopeptides and the related xenortides [52,53]. We could annotate two rhabdopeptide-related motifs with frequencies of 231 and 186 (3.7% and 3.0% of nodes, respectively). Compared to the structurally less diverse xentrivalpeptides [54] which the Mass2Motif had a frequency of 28, corresponding to 0.45% of the nodes, we can conclude that rhabdopeptide-related molecules are widespread in the *Xenorhabdus*/*Photorhabdus* extracts. The PAX peptides constitute another well-known *Xenorhabdus*/*Photorhabdus* lysine-rich peptide class [55]. The corresponding MF consisted of 13 members; indeed, they shared a Mass2Motif related to lysine (lys) and lys–lys fragments. Similarly, a leucine-leucine Mass2Motif was found in molecules annotated as xenobovid. This motif occurred in 110/6228 (1.8%) nodes pointing to several peptidic families that contain this amino acid motif—in contrast to the lys–lys amino acid motif that is very wide-spread in *Xenorhabdus*/*Photorhabdus* molecules, being present in 1500 (24%) nodes. In total, using the MolNetEnhancer workflow we could annotate 32 peptidic motifs of which we could link 11 to peptides known to be produced by *Xenorhabdus*/*Photorhabdus* strains whilst the other 21 Mass2Motifs represent substructures not yet elucidated. The peptidic nature of these Mass2Motifs was assessed by recognition of typical fragment ion patterns as seen for known peptides as well as doubly charged precursor ions that are often a sign of peptides in these extracts.

With the help of the integrative display of DEREPLICATOR and VarQuest annotation results, we could also annotate two xenoamicin-related peptidic MFs (Figure 8a,b). Xenoamicins are known to be produced by *Xenorhabdus* and eight variants have been described in detail with variants A and B present in peptidic databases [56]. Xenoamicin is a cyclic peptide consisting of a peptidic ring and peptidic tail (see Figure 8d). Interestingly, in one of the annotated MFs, not one but two Mass2Motifs were shared between most of its members (see Figure 8a). With help of DEREPLICATOR-predicted annotations of the fragment ions, we could annotate the Mass2Motif shared by almost the entire MF as being related to the xenoamicin A peptidic ring, whereas the other more abundant Mass2Motif was related to the xenoamicin peptidic tail (Figure 8c, and Appendix A). These Mass2Motifs are quite specific as we observed that 9 and 6 mass fragments, respectively, were consistently present in more than 75% of the molecular features to which the ring and tail Mass2Motifs were linked. A third Mass2Motif could be putatively annotated as xenoamicin B peptidic ring-related as its masses are +14 Da as compared to the ring A motif and xenoamicin B differs from A with an isobutyl replacing an isopropyl group. Based on the Mass2Motif presence/absence analysis in the larger MF of 32 members, we observe that 4 have links (overlap score > 0.3) to both ring A and tail motifs, 10 just have the ring A motif, three have only links to the peptidic tail motif, two share both ring A and putative ring B together with the tail Mass2Motif, and two share the putative ring B with the tail Mass2Motif (Figure 8a). Thus, this indicates how MolNetEnhancer increases the resolution in molecular networks by highlighting structural differences in between MF members.

We could also find additional MFs and singletons in which the xenoamicin ring or tail Mass2Motif was present, pointing to related peptidic molecules not linked through the modified cosine score. Further inspection with help of VarQuest annotations strengthened these annotations as VarQuest annotated modified amino acids in both rings (Figure 8, Appendix A) and the tail region (Appendix A) of xenoamicin many of which, to our knowledge, have not been reported yet, such as the one highlighted in Figure 8d where the ring-proline is likely methylated (the ring A motif is not linked to this molecular feature). In fact, xenoamicin A was annotated as variant from xenoamicin B (Appendix A) where the modified amino acid (demethylation) corresponds to previous literature findings [56], further increasing our trust in these *in silico* approaches. The smaller MF of 22 nodes consisted of doubly-charged precursor ions where no ring-related Mass2Motifs were assigned. Some members like xenoamicin A appeared in both MFs as singly and doubly charged precursor ions; the differences in motif distributions between the two MFs indicates that the initial charge has an impact on the fragmentation pathways and thus the acquired spectra given that we know the ring A is part of xenoamicin A.

Altogether, this example highlights how the MolNetEnhancer approach facilitates fragmentation based metabolomics analysis workflows by increasing the “structural resolution”, the discovery of more xenoamicin variants than previously described, and highlighting previously unseen connections between MFs and molecules. Furthermore, the integrative approach enabled straightforward annotation of Mass2Motifs found in the xenoamicin MF by using the VarQuest fragment ion annotations as guide for Mass2Motif feature annotation. Both Mass2Motif and VarQuest results strengthened each other since when predicted amino acid changes occurred in the peptidic ring, the corresponding ring-related Mass2Motif was absent, and vice versa—made possible by combining the outputs of several *in silico* tools together.

## 4. Discussion

Although significant advances have been made in molecular mining workflows, chemical annotation as well as classification tools [1,2,3,4,7,8,10,14,15,16], chemical structural annotation remains the major and most challenging bottleneck in mass spectrometry-based metabolomics as most of our biological interpretations rely on annotated structures [8,26,57]. MolNetEnhancer is a workflow that combines chemical structural information retrieved from different *in silico* tools, thus increasing structural information retrieved and enhancing biological interpretation. Here, we have chosen a representative number of *in silico* tools covering mining, annotation, and chemical annotation to provide the user with different chemical insights. Although we used DEREPLICATOR and NAP to exemplify *in silico* annotation tools here, MolNetEnhancer is platform independent, meaning that chemical structures retrieved from any *in silico* annotation platform could be used given the molecular feature identities correspond across all molecular mining and annotation tools.

Particularly in natural products research, the rapid annotation of known (i.e., dereplication) as well as unknown specialized metabolites from complex metabolic mixtures hinders interpretation in an ecological, agricultural or pharmaceutical context. Many specialized metabolites from natural sources are used as pharmaceuticals [58], in agriculture [59], or nutrition [60]; however, their discovery is inherently slow due to the above-mentioned limitations. To highlight how MolNetEnhancer can accelerate chemical structural annotation in complex metabolic mixtures from natural sources, we exemplified its use on four plant and bacterial datasets.

In the plant genus *Euphorbia*, we were able to retrieve chemical structural information of previously described pharmaceutically highly valuable diterpenoid skeletons corresponding to an annotation level 3 according to the Metabolomics Standard Initiative’s reporting standards [43]. The use of different tools combined in one data format with MolNetEnhancer allowed both for the retrieval of complementary information as well as the reinforcement of putative annotations, in cases where two independent tools pointed to the same chemical structural conclusion. Used separately, none of the tools were able to retrieve as much chemical structural information as when combined in MolNetEnhancer. Likewise, MolNetEnhancer allowed for the annotation of triterpenoids chemistries with several distinct phenolic acid modifications (e.g., vanillate, protocatechuate) in the plant family Rhamnaceae. In *Salinispora* and *Streptomyces* bacterial extracts, MolNetEnhancer aided the annotation of a previously unreported tryptophan-based MF, and a xenoamycin-related MF in the Gammaproteobacteria of the genus *Xenorhabdus* and *Photorhabdus* could be studied in more detail than in previous studies.

It is of utmost importance to note that results retrieved from MolNetEnhancer summarize results retrieved from third-party software and manual inspection and validation of all structural hypotheses remain essential. However, MolNetEnhancer significantly aids the manual inspection and validation process conducted by the expert, by making substructural as well as chemical class information readily available and visible within one data resource. As exemplified in the case studies, MolNetEnhancer can for example help in prioritizing molecular families within a molecular network, which consists of many hundreds to thousands of molecular features, be it by highlighting different chemical classes of interest or molecular families, for which only very few structural hypotheses could be retrieved, potentially highlighting novel chemistry.

Limitations introduced through data acquisition on different mass spectrometric instrument types do also apply to MolNetEnhancer. Acquiring data on different instruments can cause different MS2 fragmentation patterns, thus in some cases leading to different structural hypotheses through library matching or *in silico* structure prediction [61]. Also, the presence of low quality and/or chimeric MS2 spectra is a challenge for mass spectrometry annotation tools as the one described here, and methods that are capable of filtering-out these spectra before proceeding with *in silico* annotation tools will improve our confidence in *in silico* spectral annotation [62].

These limitations highlight the importance of good practices during data acquisition and processing to minimize the time spent analyzing mass spectrometry artefacts and improving the confidence in any downstream annotations. Here, the use of feature-based molecular networking could also help to focus the analysis on those molecular features that are very likely molecular ions [63]—and it has the added benefit that MS1 differential abundance information from LC–MS peak picking is available on the molecular features as well.

Apart from limitations caused by experimental conditions, analysis bias can be introduced for structural predictions based on chemical structures available in public databases, which are still limited especially for particular compound classes. This is particularly true for the chemical class annotations provided through ClassyFire, which rely on collecting correct or structurally closely related candidate structures from compound databases. The chemical annotation score was implemented to guide the researcher in assessing how consistent the chemical annotations are and for how many molecular features at least one candidate structure is found. The peptidic annotations by DEREPLICATOR and VarQuest come with scores, *p*-values, and false discovery rates to assess confidence in the annotations. Using MolNetEnhancer, it is now also possible to explore the consistency in peptidic annotations within MFs, along with their associated Mass2Motifs, which also assist in improving confidence in the annotations, as we have shown for the xenoamicin MFs in the nematode symbiont bacteria where the majority of the MFs were annotated with xenoamicin variants.

One limitation of the use of MS2LDA on the bacterial datasets is that most noncyclic peptidic molecular families do not share any motifs as typically analogues differ by modifications such as methylation or hydroxylation causing a shift in *m/z* in most of their mass fragment peaks. Incorporation of amino acid-related mass differences as features for MS2LDA could be a route to also discover Mass2Motifs for noncyclic peptides. As it is, cyclic peptides do often contain one or more Mass2Motifs and peptides containing positively charged amino acids such as lysine and leucine have this structural information represented by Mass2Motifs. Furthermore, many Mass2Motifs are currently still unannotated, which hampers fast structural analysis. To partially solve this bottleneck, MotifDB (www.ms2lda.org/motifdb) was recently introduced [64] and the here annotated Mass2Motif sets from the four case studies are made available through MotifDB for matching against Mass2Motifs found in other MS2LDA experiments. Furthermore, this will allow to use a combination of “supervised” (annotated) Mass2Motifs and “unsupervised” (free) Mass2Motifs in future MS2LDA experiments on data of related samples thereby accelerating structural annotation since part of the motifs already discovered do not need to be reannotated.

Despite the limitations discussed above, MolNetEnhancer assists in metabolite annotations by its combined analysis of chemical class annotations, structural annotations, and Mass2Motif annotations. If these annotations support each other, as for example for the actinomycin MF in the marine sediment bacteria, there is more confidence that these in silico annotations will indeed be correct. It is noteworthy that the modularity of MolNetEnhancer allows for complementary sources of structural information to be added on in future. We showed that MolNetEnhancer is a practical tool to annotate the chemical space of complex metabolic mixtures using a panel of complementary in silico annotation tools for mass spectrometry based metabolomics experiments. Although we have highlighted the use of MolNetEnhancer using two plant and bacterial datasets, MolNetEnhancer is sample type-independent and may be used for any mass spectrometry-based metabolomics experiment, where chemical structural annotation and interpretation is of interest. Future work will focus on making the complete MolNetEnhancer workflow available within the GNPS platform in order to further increase its user friendliness. Currently, the chemical classification workflow is available to run within the GNPS framework directly outputting an annotated network (see URL in code availability Section 7). Furthermore, the integration of other existing and future metabolome mining and annotation tools in the output of MolNetEnhancer is also planned to extend on the initial set of in silico tools that it currently can combine.

## 5. Conclusions

MolNetEnhancer is a powerful tool to accelerate chemical structural annotation within complex metabolic mixtures through the combined use of mass spectral molecular networking, substructure discovery, in silico annotation as well as chemical classifications provided by ClassyFire. The MolNetEnhancer workflow is presented both as an open source Python module and R package, allowing easy access and usability by the community as well as the possibility for customization and further development by integration into future collaborative modular tools and by integration of other existing or future metabolome mining and annotation tools. Whilst its use was showcased using natural product examples, we expect that MolNetEnhancer will also enhance biological and chemical interpretations in other scientific fields such as clinical and environmental metabolomics.

## 6. Data Availability

Publicly available mass spectrometry fragmentation data sets from four studies were used for this study. Details on how samples and data were collected can be found in the original studies [27,28,48,50]. Here, we list links to the different analyses that were done on each of the studies. Through these links, all used settings and parameters can be retrieved.

Data from case studies 1 & 2 illustrating MolNetEnhancer applied to feature-based molecular networking are publicly accessible through the links listed below.

Case study 1: *Euphorbia* study—combined analysis of 43 *Euphorbia* plant extracts

MASSIVE: MSV000081082 https://massive.ucsd.edu/ProteoSAFe/dataset.jsp?task=c9f09d31a24c475e87a0a11f6e8889e7GNPS Molecular Networking job: https://gnps.ucsd.edu/ProteoSAFe/status.jsp?task=26326c233918419f8dc80e8af984cdaeGNPS NAP jobs: https://proteomics2.ucsd.edu/ProteoSAFe/status.jsp?task=2cfddd3b8b1e469181a13e7d3a867a6f and https://proteomics2.ucsd.edu/ProteoSAFe/status.jsp?task=184a80db74334668ae1d0c0f852cb77cMS2LDA experiment: http://ms2lda.org/basicviz/summary/390

Case study 2: Rhamnaceae study—combined analysis of 71 Rhamnaceae plant extracts

MASSIVE: MSV000081805 https://massive.ucsd.edu/ProteoSAFe/dataset.jsp?task=36f154d1c3844d31b9732fbaa72e9284GNPS Molecular Networking job: https://gnps.ucsd.edu/ProteoSAFe/status.jsp?task=e9e02c0ba3db473a9b1ddd36da72859bGNPS NAP job: https://proteomics2.ucsd.edu/ProteoSAFe/status.jsp?task=6b515b235e0e4c76ba539524c8b4c6d8MS2LDA experiment: http://ms2lda.org/basicviz/summary/566

GNPS example study used in Jupyter notebook to show MolNetEnhancer based on feature-based molecular networking—subset of American Gut Project:MASSIVE: MSV000082678 https://massive.ucsd.edu/ProteoSAFe/dataset.jsp?task=de2d18fd91804785bce8c225cc94a44GNPS Molecular Networking job: https://gnps.ucsd.edu/ProteoSAFe/status.jsp?task=b817262cb6114e7295fee4f73b22a3adGNPS NAP job: https://proteomics2.ucsd.edu/ProteoSAFe/status.jsp?task=c4bb6b8be9e14bdebe87c6ef3abe11f6MS2LDA experiment: http://ms2lda.org/basicviz/summary/907

Data from case studies 3 & 4 illustrating MolNetEnhancer applied to classical molecular networking are publicly accessible through the links listed below.

Case study 3: Marine-sediment bacteria study—combined analysis of 120 *Salinospora* and 26 *Streptomyces* bacterial strain extracts
MASSIVE: MSV000078836, MSV000078839 https://massive.ucsd.edu/ProteoSAFe/dataset.jsp?task=9277186021274990a5e646874a435c0d
https://massive.ucsd.edu/ProteoSAFe/dataset.jsp?task=a507232a787243a5afd69a6c6fa1e508GNPS Molecular Networking job: http://gnps.ucsd.edu/ProteoSAFe/status.jsp?task=c36f90ba29fe44c18e96db802de0c6b9GNPS NAP job: https://proteomics2.ucsd.edu/ProteoSAFe/status.jsp?task=60925078e0c148cbaba3593569e983d6GNPS DEREPLICATOR 0.005 job: https://gnps.ucsd.edu/ProteoSAFe/status.jsp?task=0ad6535e34d449788f297e712f43068aGNPS DEREPLICATOR 0.05 job: https://gnps.ucsd.edu/ProteoSAFe/status.jsp?task=e494a63be6d34747a4b8cdfb838ef96eGNPS VARQUEST 0.005 job: https://gnps.ucsd.edu/ProteoSAFe/status.jsp?task=f1f00c1c20ba4f61ad471d340066df76GNPS VARQUEST 0.05 job: https://gnps.ucsd.edu/ProteoSAFe/status.jsp?task=f5ffcc8f63ab4e6f96a97caabc11048bMS2LDA annotation experiment: http://ms2lda.org/basicviz/summary/551MS2LDA MolNetEnhancer workflow experiment: http://ms2lda.org/basicviz/summary/912

Case study 4: Nematode symbionts study—combined analysis of 25 *Xenorhabdus* and 5 *Photorhabdus* bacterial strain extracts

MASSIVE: MSV000081063 https://massive.ucsd.edu/ProteoSAFe/dataset.jsp?task=dcc30b777c344d668a5626d01f26c9a0GNPS Molecular Networking job: https://gnps.ucsd.edu/ProteoSAFe/status.jsp?task=aaff4721951b4d92b54ecbd2fe4b9b4fGNPS NAP job: http://gnps.ucsd.edu/ProteoSAFe/status.jsp?task=677f076eb04b4518958ca8cd56b4c753GNPS DEREPLICATOR 0.005 job: http://gnps.ucsd.edu/ProteoSAFe/status.jsp?task=338b422483d1432e82afd1bf848f1292GNPS DEREPLICATOR 0.05 job: http://gnps.ucsd.edu/ProteoSAFe/status.jsp?task=83bca3c45665470891d41ead275dcae7GNPS VARQUEST 0.005 job: http://gnps.ucsd.edu/ProteoSAFe/status.jsp?task=20cfb9af4a244feea102aa9c9da2651cGNPS VARQUEST 0.05 job: http://gnps.ucsd.edu/ProteoSAFe/status.jsp?task=a4ffda169823476a9b1e81616aeccbdaMS2LDA annotation experiment: http://ms2lda.org/basicviz/summary/570MS2LDA MolNetEnhancer workflow experiment: http://ms2lda.org/basicviz/summary/917

GNPS example study used in Jupyter notebook to show MolNetEnhancer based on classical molecular networking—drug metabolism in set of sputum samples:MASSIVE: MSV000081098 https://gnps.ucsd.edu/ProteoSAFe/result.jsp?task=7c4b25d21a6348df9a6942d3071a4b1f&view=advanced_viewGNPS Molecular Networking job: https://gnps.ucsd.edu/ProteoSAFe/status.jsp?task=b76dd5a123e54a7eb42765499f9163a5GNPS NAP job: https://proteomics2.ucsd.edu/ProteoSAFe/status.jsp?task=cb63770fe307410492468f62f9edb8f3VarQuest job: https://gnps.ucsd.edu/ProteoSAFe/status.jsp?task=4d971b8162644e869a68faa35f01b915DEREPLICATOR job: https://gnps.ucsd.edu/ProteoSAFe/status.jsp?task=c62d3283752f4f98b1720d0a6d1ee65bMS2LDA experiment: http://ms2lda.org/basicviz/summary/909

## 7. Code Availability

The MolNetEnhancer package in R including Jupyter notebooks with an exemplary analysis workflow for mapping Mass2Motifs and chemical class annotations onto classical and feature-based molecular networks is publicly accessible at https://github.com/madeleineernst/RMolNetEnhancer and the MolNetEnhancer package in Python including Jupyter notebooks with an exemplary analysis workflow for mapping Mass2Motifs and chemical class annotations onto classical and feature-based molecular networks is publicly accessible at https://github.com/madeleineernst/pyMolNetEnhancer. A beta version of the MolNetEnhancer workflow is also available from within GNPS: https://gnps.ucsd.edu/ProteoSAFe/index.jsp?params=%7B%22workflow%22:%22MOLNETENHANCER%22%7D. This currently outputs the chemical class annotated molecular network by user provided task ids to the individual jobs run within GNPS.

## 8. Tutorials

Tutorials to get familiar with individual tools from which the output is combined with MolNetEnhancer can be found here.

GNPS molecular networking: 
https://ccms-ucsd.github.io/GNPSDocumentation/networking
DEREPLICATOR/VarQuest: 
https://ccms-ucsd.github.io/GNPSDocumentation/dereplicator
Network annotation propagation: 
https://ccms-ucsd.github.io/GNPSDocumentation/nap
ClassyFire: 
http://classyfire.wishartlab.com
MS2LDA: 
https://ccms-ucsd.github.io/GNPSDocumentation/ms2lda/

http://ms2lda.org/user_guide
MolNetEnhancer workflow tutorials in both R and Python can be found here: 
https://github.com/madeleineernst/pyMolNetEnhancer

https://github.com/madeleineernst/RMolNetEnhancer


## Figures and Tables

**Figure 1 metabolites-09-00144-f001:**
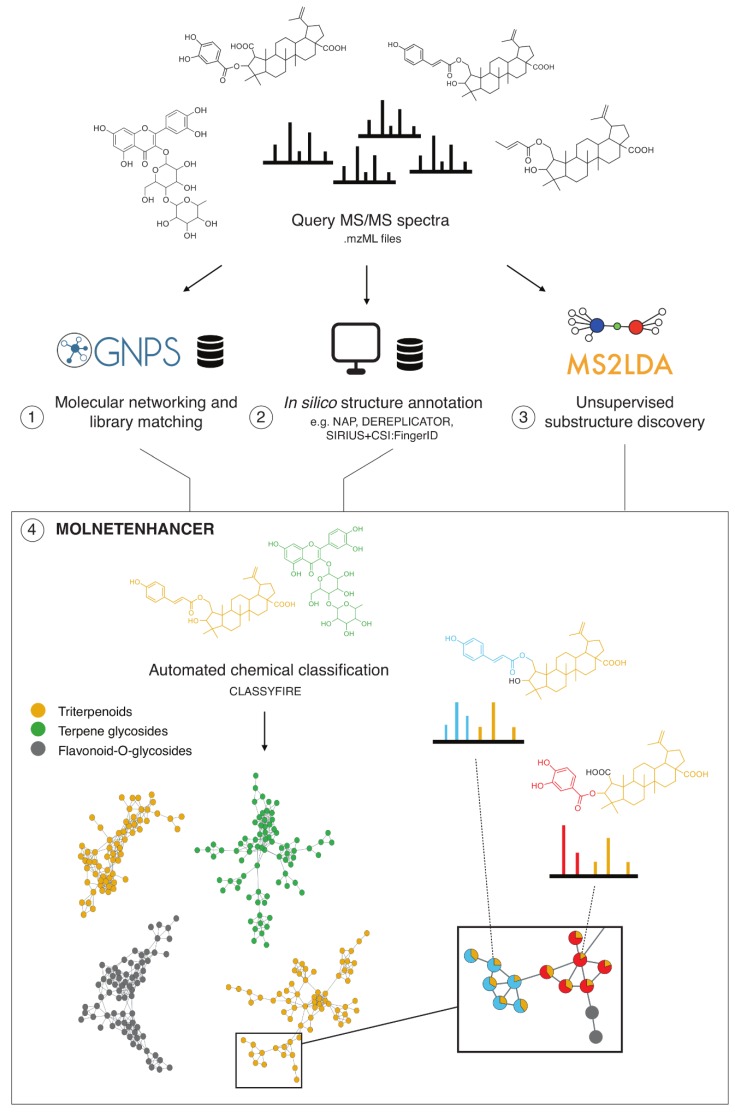
Schematic overview of the MolNetEnhancer workflow. Starting with mass spectrometry data in the mzML format obtained from complex metabolic mixtures the user creates (**1**) mass spectral molecular networks in GNPS, (**2**) performs in silico structure annotation (e.g., through NAP, DEREPLICATOR or SIRIUS+CSI:FingerID), and (**3**) performs unsupervised substructure discovery through MS2LDA. Steps 1–3 are performed prior to the MolNetEnhancer workflow within the respective platforms. MolNetEnhancer is then used in (**4**) to map information layers obtained from all three platforms independently on top of each other resulting in network-wide chemical class information and more detailed substructure information within molecular families (as exemplified for the organic acid conjugates in the enlarged part of the triterpenoid molecular family on the right).

**Figure 2 metabolites-09-00144-f002:**
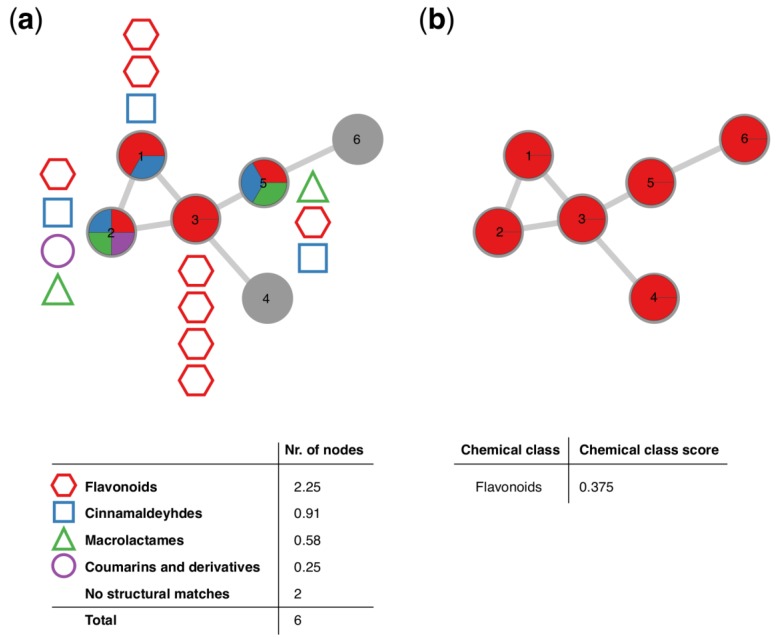
Schematic overview of how the chemical classification score is calculated and visualized within a molecular family. (**a**) Schematic overview of hypothetical structural annotations within a molecular family consisting of 6 nodes. Out of the 6 nodes, chemical structural information could be retrieved for 4, where each node can consist of structural annotations to multiple different (e.g., node 2) or identical (e.g., node 3) chemical classes. The total number of nodes per chemical class retrieved is calculated and the most abundant chemical class is assigned to the molecular family, resulting in (**b**). Schematic overview of the molecular family shown in (**a**), classified as ‘flavonoids’ at the chemical class level by MolNetEnhancer, with a score of 0.375, translating to the majority of the putative structural annotations within this molecular family (2.25) belong to the flavonoid structural class.

**Figure 3 metabolites-09-00144-f003:**
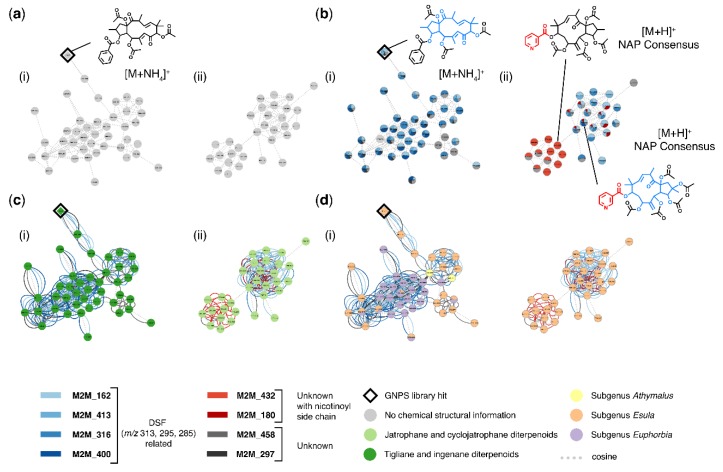
MolNetEnhancer increases chemical structural information obtained for *Euphorbia* specialized metabolites. (**a**) Mass spectral molecular network showing two molecular families of *Euphorbia* specialized metabolites. Using GNPS library matching only one molecular feature could be putatively annotated. Manual annotation propagation is limited for family (i) and impossiblefor family (ii). (**b**) Using MolNetEnhancer, substructural Mass2Motifs can be visualized within the network; both molecular family (i) and (ii) contain Mass2Motifs related to a *Euphorbia* diterpene spectral fingerprint (DSF) and molecular family (ii) contains Mass2Motifs related to a nicotinoyl side chain. Mass2Motifs are mapped on the nodes as pie charts with an area proportional to their overlap score, a score measuring how much of the Mass2Motif is present in the spectrum, whereas dotted lines connecting the nodes represent features with a MS2 spectral similarity of a cosine score over 0.6 (**c**) Most chemical structures retrieved for molecular family (i) and (ii) are diterpenoids of the jatrophane, tigliane or ingenane type, which both can result in a DSF with *m*/*z* 313, 295, or 285. Substructures with mass fragments characteristic of these *Euphorbia* DSFs were also found within the Mass2Motifs. Node colors represent most abundant chemical classes, colored lines connecting the nodes represent shared Mass2Motifs, and dotted lines connecting the nodes represent features with a MS2 spectral similarity of a cosine score over 0.6 (**d**) *Euphorbia* diterpenoid skeletons of the jatrophane, deoxy tigliane, or ingenane ester type are found within all *Euphorbia* subgeneric clades, whereas nicotinoyl sidechain modifications are unique to subgenus *Esula*. Node colors represent summed peak area per *Euphorbia* subgeneric clade, colored lines connecting the nodes represent shared Mass2Motifs, and dotted lines connecting the nodes represent features with a MS2 spectral similarity of a cosine score over 0.6.

**Figure 4 metabolites-09-00144-f004:**
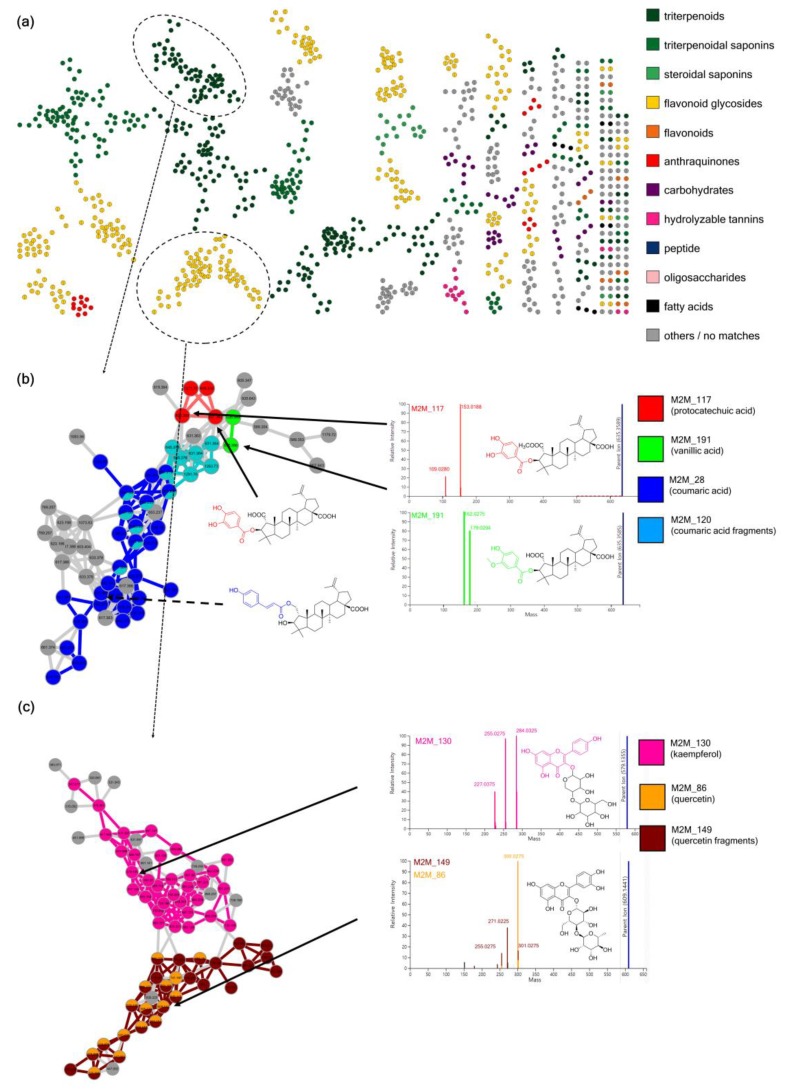
MolNetEnhancer increases chemical structural information obtained for Rhamnaceae specialized metabolites. (**a**) Structural annotation for molecular families was suggested based on consensus-based classification of NAP in silico structure annotation. (**b**) Subtle chemical differences of phenolic acid moieties can be visualized within the molecular family of triterpenoid esters based on Mass2Motifs. (**c**) Molecular family annotated as flavonoid glycosides reveals two subfamilies by Mass2Motif mapping: the pink Mass2Motif is related to the kaempferol core structure, whereas the orange and brown Mass2Motifs are related to the quercetin core structure—two related yet distinct flavonoid structures.

**Figure 5 metabolites-09-00144-f005:**
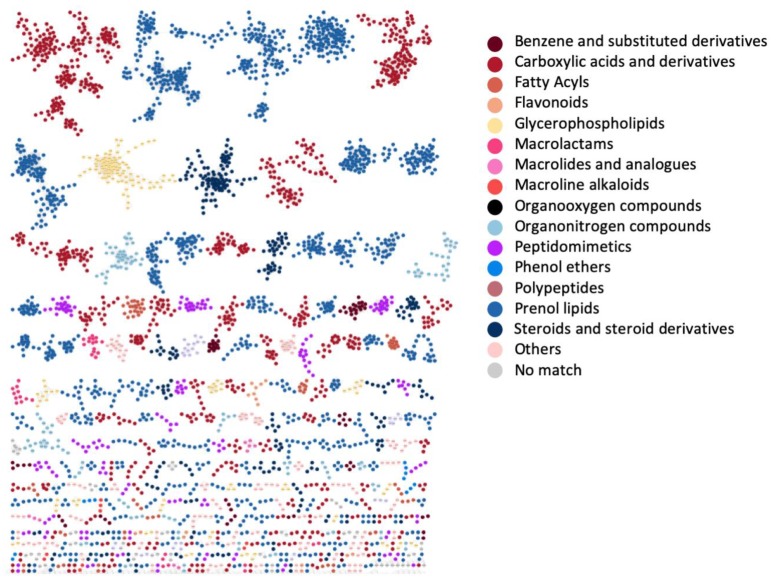
Marine sediment *Salinispora*/*Streptomyces* molecular network colored by 15 selected chemical class terms as indicated in the legend. In total, 50 different class terms were annotated in the network using MolNetEnhancer, indicating that the metabolic output of the *Salinispora*/*Streptomyces* strains is chemically very diverse. We can observe that the larger molecular families are mostly annotated with prenol lipids (blue) and carboxylic acids and derivatives (red). Furthermore, for a couple of MFs no chemical class annotations were obtained as no candidate structures were retrieved through any of the annotation tools.

**Figure 6 metabolites-09-00144-f006:**
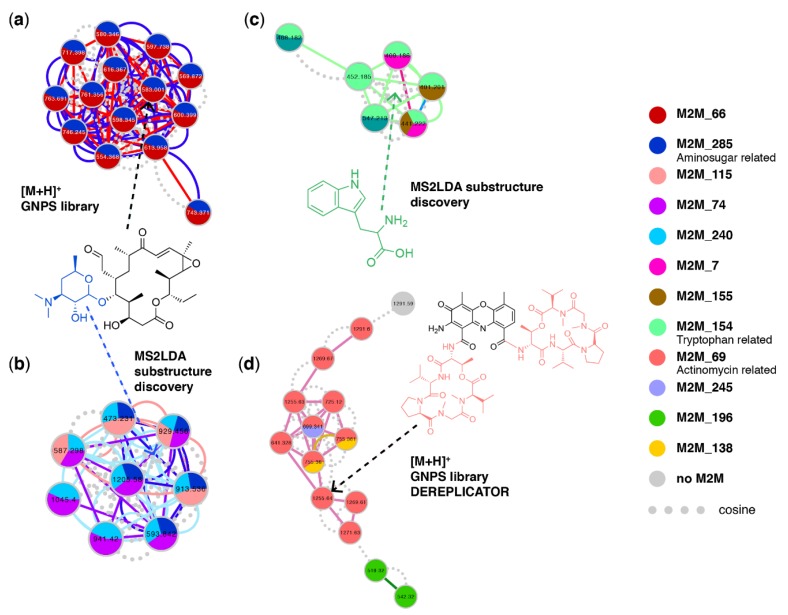
Molecular families from marine sediment bacteria with color coded Mass2Motif substructure information mapped on them, with (**a**) rosamicin-related molecular family found through GNPS library hits where all members contain an amino sugar-related motif as colored in blue in its depicted structure—substructures or motifs found within each molecular feature are mapped on the nodes as pie charts, where the relative abundance of each motif represents the overlap score, a score measuring how much of the motif is present in the spectrum. Furthermore, motifs shared between two nodes are visualized as colored continuous lines (edges) connecting the nodes whereas dashed lines (edges) represent a cosine score of over 0.6, (**b**) Yet unknown molecular family that shares an amino sugar-related motif connecting this MF to (**a**) by sharing a substructure, (**c**) tryptophan-related molecular family sharing the Tryptophan Mass2Motif, and (**d**) actinomycin-related molecular family—found through GNPS library hits and further validated with help of DEREPLICATOR results—sharing an Actinomycin related motif across most of its members. The actinomycin D (Daptomycin) structure is depicted with the Mass2Motif substructure highlighted in color: the peptide lactone ring present twice in the molecule. In all MFs, nodes are colored based on Mass2Motif overlap scores and the edges show if cosine score-connected nodes share similar Mass2Motifs. It can be seen that in all families multiple motifs are shared across some of its members.

**Figure 7 metabolites-09-00144-f007:**
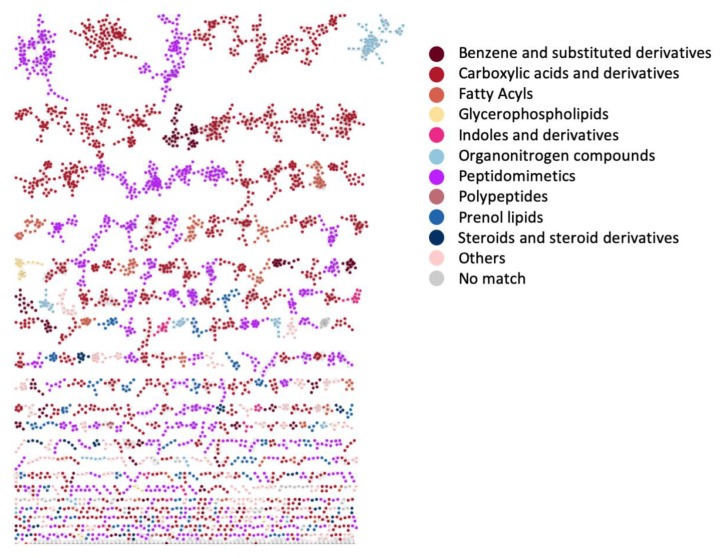
Nematode symbionts *Photorhabdus*/*Xenorhabdus* network colored by 10 selected chemical class terms as indicated in the legend. In total, 49 different class terms were annotated in the network using MolNetEnhancer. We can observe that the larger molecular families as well as many smaller molecular families are mostly annotated with peptidomimetics (purple) and carboxylic acids and derivatives (red). This is consistent with earlier findings that these nematode symbionts produce a wide array of peptidic products.

**Figure 8 metabolites-09-00144-f008:**
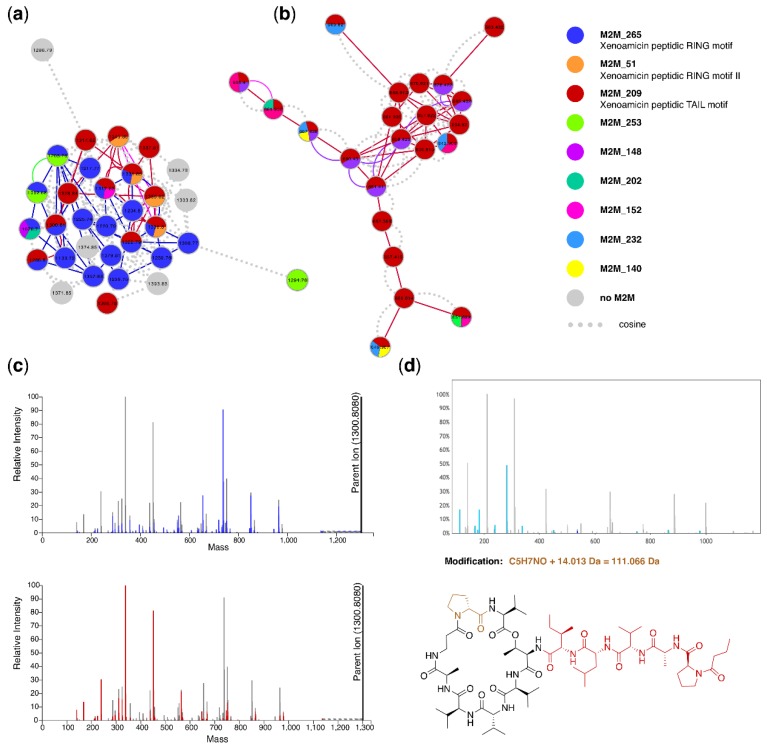
Xenoamicin-related molecular families annotated by MolNetEnhancer with (**a**) MF of 32 nodes of which 23 were annotated with at least one xenoamicin modified structure (xenoamicin A or B) by either VarQuest or DEREPLICATOR with VarQuest using 0.005 Da fragment binning assigning most xenoamicin structures (FDRs mostly < 2.5). This MF also contains nodes sharing all Mass2Motifs related to xenoamicin structures with two ring and tail-related Mass2Motifs. Mass2Motif 265 contains mass fragments related to xenoamicin A, whereas masses in Mass2Motif 51 are shifted with 14 Da pointing towards xenoamicin B. The MF consists of singly charged molecular features. (**b**) Related MF of which 20 out of 22 nodes were annotated with xenoamicin modified structures (FDRs mostly < 2.5). This MF only shares the Mass2Motif annotated as xenoamicin tail-related and consists of doubly-charged precursor ions. (**c**) Xenoamicin A spectrum in the ms2lda.org environment with (top) ring-related Mass2Motif highlighted and (bottom) tail-related Mass2Motif highlighted with the corresponding blue and red colors as in (**a**) and (**b**). (**d**) VarQuest annotation of xenoamicin modified peptide where a ring proline indicated in brown is likely methylated. All light blue peaks in the mass spectrum were annotated by VarQuest. The red part in the xenoamicin structure corresponds to the selected fragment of *m/z* 537.348, which includes the tail part, whereas the light blue amino acid is annotated to be modified with a mass shift of 14.013 Da that likely corresponds to a methylation. Indeed, the Mass2Motif related to the xenoamicin tail is found in this fragmentation spectrum, whereas the ring Mass2Motif is absent.

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
