# Peer review of "MolNetEnhancer: Enhanced Molecular Networks by Integrating Metabolome Mining and Annotation Tools"

_metabolites, 2019, doi:10.3390/metabo9070144_

Round 1

Reviewer 1 Report

The paper describes a new tool, available both in Python and in R, that is able to combine information from several tools into information rich molecular networks. The concept is great. It lowers the learning curve and the required programming skills to make advanced network analysis of metabolomics studies and aids identification by providing easy visualizations of summarized information. I look forward to using this tool.

Though I find the work important and of high value I find that the paper itself is structured in a way that makes it very inaccessible to the target audience. The paper puts an unreasonable expectation of prior familiarity with the involved tools on the reader. The tools involved are all relatively new and often conceptually less than straight-forward. You can therefore not expect the reader to be familiar with the individual tools. Referring just to the information in this paper I therefore find myself with only a superficial understanding of the workflow and the output graphs even if I should be part of the target audience for this tool.

I feel a major obstacle for the understanding is the macro-structure of the paper (The Intro --> Results --> Discussion --> M&M --> Conclusion). With the M&M at the end the breakdown of the individual steps gets lost. I feel this paper would benefit tremendously by some reorganization. I know this is the order described in the instructions to authors but other papers have used different structures (e.g. https://doi.org/10.3390/metabo9050103). I would suggest a M&M section that goes before the results and discussion that is written is as much of a tutorial style as possible. With the individual steps clearly outlined first you have a better chance of understanding the results. I would then suggest some of the explanations of the tools go into the results to explain exactly what you have gotten in each step. The figure texts are exceedingly long and I believe a lot of the explanation from here could be used when you describe each step in the main text.  

Another important point that only occurred to me after getting to the M&M section and the online tutorial is that you need to run many of the analysis tools manually on web services. Reading the paper I assumed that the R and Python packages would query these tools internally. It should be clear that (as far as I understood) the function of the package is to combine the output from different tools into enriched network graphs. Not to provide a single entryway to all the required tools. This heavily limits the usability in workflows when much of the work is manual.

Other suggestions

Figure 1 needs a lot of work to fulfill its purpose. For instance: in the text MS2LDA is the first step. Yet it is to the right in the second "row". You should help the reader by having a figure you can follow while you read that goes as much as possible left to right and top to bottom. Worse is that a lot of the tools mentioned (Classyfire, NAP, DEREPLICATOR etc) is nowhere in the figure. Please help the reader!

You could perhaps use the RCytoscape to prepare the graphs instead of having to manually having to import and select data in Cytoscape.

The R package appears incomplete. The paper itself lists that it has less features but it appears that it consist of a single function with less than 100 lines of code. Even the description file is a template file not fully complete.

Since the Python package has more features perhaps you could use the reticulate package to make the functionality available in R. You would then perhaps also avoid having to maintain separate code.

Page 4: first two paragraphs contain citations in a different format.

In the M&M section I would move all the links to the processing tools to supplementary.

At the end of case study 2 in the M&M you end by "listed below:". But there seems to be nothing below.

"Annotation" is misspelled in figure 1.

"cinnamaldehyde" is misspelled in figure 8.

Author Response

We thank the reviewers for their constructive feedback and suggestions. We have taken on board as much as we could in particular to provide more explanation on the different tools the output of which MolNetEnhancer can combine to aid in structural annotations. Whilst we do not have the space to provide detailed information on the use of all the tools and each of the tools was published in separate papers to which we refer to, we have included sentences in the introduction and methods section on what the tools do and the expected output is and how these could contribute to metabolite annotation and to MolNetEnhancer. We have also provided links to tutorials for those tools that are available to guide the reader in their use. The discussion highlights strengths and limitations of the workflow which we deem important to communicate with the reader. Based on reviewer 1’s suggestion we have also moved the M&M part prior to the Results part deviating from the template order. Please find a detailed point-by-point answer below.

Reviewer 1

The paper describes a new tool, available both in Python and in R, that is able to combine information from several tools into information rich molecular networks. The concept is great. It lowers the learning curve and the required programming skills to make advanced network analysis of metabolomics studies and aids identification by providing easy visualizations of summarized information. I look forward to using this tool.

We thank the reviewer for his positive words and enthusiasm about our work.

Though I find the work important and of high value I find that the paper itself is structured in a way that makes it very inaccessible to the target audience. The paper puts an unreasonable expectation of prior familiarity with the involved tools on the reader. The tools involved are all relatively new and often conceptually less than straight-forward. You can therefore not expect the reader to be familiar with the individual tools. Referring just to the information in this paper I therefore find myself with only a superficial understanding of the workflow and the output graphs even if I should be part of the target audience for this tool.

This comment relates to the comment below - as we agree that it should be clear what the different tools do and how they can work together we have added text on this in the manuscript. We have also provided links to relevant tutorials as these are available to get familiar with these tools. We hope this will indeed higher the impact of combining the outputs in a unified network. 

I feel a major obstacle for the understanding is the macro-structure of the paper (The Intro --> Results --> Discussion --> M&M --> Conclusion). With the M&M at the end the breakdown of the individual steps gets lost. I feel this paper would benefit tremendously by some reorganization. I know this is the order described in the instructions to authors but other papers have used different structures (e.g. https://doi.org/10.3390/metabo9050103). I would suggest a M&M section that goes before the results and discussion that is written is as much of a tutorial style as possible. With the individual steps clearly outlined first you have a better chance of understanding the results. I would then suggest some of the explanations of the tools go into the results to explain exactly what you have gotten in each step. The figure texts are exceedingly long and I believe a lot of the explanation from here could be used when you describe each step in the main text.  

Following the reviewer’s suggestion, we have moved most of the Material and Methods up to after the Introduction and separated it from newly formed Data availability, Code availability, and Tutorials sections that we left at the end of the manuscript not to break the flow of the manuscript whilst maintaining interactive links to the analysis results. In the Discussion, we discuss the advantages and limitations of the MolNetEnhancer workflow - whereas diverse tutorials that include screenshots of the relevant steps are online available. In fact, a tutorial for MolNetEnhancer is available from the GitHub pages. (https://github.com/madeleineernst/pyMolNetEnhancer and https://github.com/madeleineernst/rMolNetEnhancer) something we clarified in the manuscript. We have also added links to tutorials for the other tools we used output from in Section 8. We expect that this will help the users to find their way through the tools. 

Another important point that only occurred to me after getting to the M&M section and the online tutorial is that you need to run many of the analysis tools manually on web services. Reading the paper I assumed that the R and Python packages would query these tools internally. It should be clear that (as far as I understood) the function of the package is to combine the output from different tools into enriched network graphs. Not to provide a single entryway to all the required tools. This heavily limits the usability in workflows when much of the work is manual.

This observation is true. MolNetEnhancer does not allow yet for querying the individual tools internally and indeed at this stage we focus on combining the outputs of several tools in a meaningful manner. However, to ease user friendliness we are currently working on implementing the MolNetEnhancer workflow within GNPS (see Discussion), where in the future, it will be possible to launch all workflows in an automated and combined manner. We have added one author, Christoper Chen, who has been instrumental to perform this integration over the last weeks. Before this, we would like to publish the underlying code with this manuscript, so users can modify and also integrate the output with other, not GNPS-based workflows - both in R and Python. We have also clarified that MolNetEnhancer unites the output of diverse mining and annotation tools throughout the manuscript text.

Other suggestions

Figure 1 needs a lot of work to fulfill its purpose. For instance: in the text MS2LDA is the first step. Yet it is to the right in the second "row". You should help the reader by having a figure you can follow while you read that goes as much as possible left to right and top to bottom. Worse is that a lot of the tools mentioned (Classyfire, NAP, DEREPLICATOR etc) is nowhere in the figure. Please help the reader!

We have reworked Figure 1 to address the reviewer’s requests, by adding numbers, which read from left to right and top to bottom to guide the reader. Also we have included explicit citation of ClassyFire, NAP and Dereplicator in the figure. Also relevant input file formats are mentioned now.

You could perhaps use the RCytoscape to prepare the graphs instead of having to manually having to import and select data in Cytoscape.vThe R package appears incomplete. The paper itself lists that it has less features but it appears that it consist of a single function with less than 100 lines of code. Even the description file is a template file not fully complete.

Since the Python package has more features perhaps you could use the reticulate package to make the functionality available in R. You would then perhaps also avoid having to maintain separate code.

We thank the reviewer for this suggestion. We have now added all functionalities available in the Python package also in R using the reticulate package. Furthermore, we have implemented a graph output also in R using the igraph package. Thus, all the features available in Python are now also implemented in the R package. Also we thank the reviewer for spotting the incomplete description file for the R package, which we now have completed.

Page 4: first two paragraphs contain citations in a different format.

We thank the reviewer for spotting this omission and have addressed it now.

In the M&M section I would move all the links to the processing tools to supplementary.

We think providing links to the relevant results sections will ease the process for users to reuse the tools on their own data using similar settings/parameters as those adopted by our study. We therefore opt to keep them inside the main manuscript, but for ease of readability we moved them to a separate section, Section 6, ‘Data availability’.

At the end of case study 2 in the M&M you end by "listed below:". But there seems to be nothing below.

The word below refers to the next section of links provided to the individual tool analysis results. The first section (Case studies 1 & 2) was done using “Feature-based molecular networking”, the second section (Case studies 3 & 4) using classical molecular networking. This has been clarified in the text now.

"Annotation" is misspelled in figure 1.

"cinnamaldehyde" is misspelled in figure 8.

We thank the reviewer for spotting these spelling errors and have amended them.

Reviewer 2 Report

In this manuscript, van der Hooft et al. describe a powerful tool to enhance the annotation of molecular networks. Furthermore, they have shown its utility on several case studies.This is a long-awaited tool by the community and I am sure that it will find tremendous applications. 

I would recommend the publication of this very nice work in Metabolites after some minor corrections.

General points: 

- To enhance the understanding of the MolNetEnhancer tool, I suggest to add a general figure that illustrates the MolNetEnhancer workflow, for e.g. Which type of file as an input which one as an output. Try to connect this tool with the existing softwares available in the molecular networking environment. I think that this will help the reader to better understand the tool.

Minor details:

page 1, Please, MetWork should be cited among the numerous annotation tools (https://academic.oup.com/bioinformatics/article/35/10/1795/5116145).

page 3, in the title 2.2 "Euphorbia" should not be italicized

page 4, the statement "biosynthetically derived from a head-to-tail cyclization of the tetraprenyl

pyrophosphate precursor" should be replaced by "biosynthetically derived from the geranyl-geranyl pyrophosphate precursor"

page 8, please correct, " Prenol lipids" by "Phenol lipids"

            I think that "amino acid lactone loop" should be replaced by "depsipeptide moiety" the same terms can be found in the caption of Figure 5 (page 10).

Author Response

We thank the reviewers for their constructive feedback and suggestions. We have taken on board as much as we could in particular to provide more explanation on the different tools the output of which MolNetEnhancer can combine to aid in structural annotations. Whilst we do not have the space to provide detailed information on the use of all the tools and each of the tools was published in separate papers to which we refer to, we have included sentences in the introduction and methods section on what the tools do and the expected output is and how these could contribute to metabolite annotation and to MolNetEnhancer. We have also provided links to tutorials for those tools that are available to guide the reader in their use. The discussion highlights strengths and limitations of the workflow which we deem important to communicate with the reader. Based on reviewer 1’s suggestion we have also moved the M&M part prior to the Results part deviating from the template order. Please find a detailed point-by-point answer below.

Reviewer 2

In this manuscript, van der Hooft et al. describe a powerful tool to enhance the annotation of molecular networks. Furthermore, they have shown its utility on several case studies.This is a long-awaited tool by the community and I am sure that it will find tremendous applications. 

I would recommend the publication of this very nice work in Metabolites after some minor corrections.

We thank the reviewer for the positive words and enthusiasm about our work.

General points: 

- To enhance the understanding of the MolNetEnhancer tool, I suggest to add a general figure that illustrates the MolNetEnhancer workflow, for e.g. Which type of file as an input which one as an output. Try to connect this tool with the existing softwares available in the molecular networking environment. I think that this will help the reader to better understand the tool.

We thank the reviewer for the constructive feedback. We have modified Figure 1 in such a way that it now illustrates the MolNetEnhancer workflow in more detail. Input files are listed and the reader is guided through the workflow by numbers reading from left to right and top to bottom. We hope that this will help the reader for a better understanding of the MolNetEnhancer workflow.

Minor details:

page 1, Please, MetWork should be cited among the numerous annotation tools (https://academic.oup.com/bioinformatics/article/35/10/1795/5116145).

The MetWorks tool as well as the MetLin platform were added to the list of numerous annotation tools available.

page 3, in the title 2.2 "Euphorbia" should not be italicized

The word Euphorbia refers to the plant genus Euphorbia. Genera names should always be italicized by taxonomic nomenclature convention. We have therefore not de-italicized Euphorbia in title 2.2.  

page 4, the statement "biosynthetically derived from a head-to-tail cyclization of the tetraprenyl

pyrophosphate precursor" should be replaced by "biosynthetically derived from the geranyl-geranyl pyrophosphate precursor"

Literature describing the biosynthesis of Euphorbia diterpenoids are most commonly referring to a head-to-tail cyclization of the tetraprenyl pyrophosphate precursor rather than a geranyl-geranyl pyrophosphate precursor (Appendino, 2016 [https://www.ncbi.nlm.nih.gov/pubmed/27380406]; Lanzotti, 2013 [https://link.springer.com/referenceworkentry/10.1007%2F978-3-642-22144-6_192]; Vasas and Hohmann, 2014). Therefore we would like to maintain the same term on page 4.

page 8, please correct, " Prenol lipids" by "Phenol lipids"

We confirm that the term we intended to use was Prenol lipids, a chemical ontology term coined by ClassyFire (http://classyfire.wishartlab.com/tax_nodes/C0000259). 

I think that "amino acid lactone loop" should be replaced by "depsipeptide moiety" the same terms can be found in the caption of Figure 5 (page 10).

We agree that it is important to be consistent and have rephrased "amino acid lactone loop" to “peptide lactone ring (depsipeptide moiety)” following the reviewer’s suggestion to make it consistent with the Figure caption and indicate it is a depsipeptide moiety.

Round 2

Reviewer 1 Report

Hats off. Well done with the rework!

Please check the R package. It fails build CHECK badly. The author field is misformed and dependencies are not declared properly. You mention some dependencies in the README but I don't see them declared in the package either.

Author Response

We thank the reviewer for testing the R package and apologise for the build CHECK error. This has been fixed now, and all dependencies are now also declared in the DESCRIPTION file of the package.

Furthermore, we corrected the affiliation of author 1 (Madeleine Ernst) by adding 'Center for Newborn Screening', the name of author 7 (Christopher Chen) by fixing a typo, and also corrected spelling mistakes in the Author Contributions Section (author Christopher Chen was wrongly listed as C.H. and C.S. which now has been corrected to C.C.)

We hope you will find the manuscript suitable for publication now and look forward to hear back.